# A Qualitative Study of Hospital Interior Environments during the COVID-19 Pandemic

**DOI:** 10.3390/ijerph20043271

**Published:** 2023-02-13

**Authors:** Suyeon Bae

**Affiliations:** Department of Housing & Interior Design, Age Tech-Convergence Major, Kyung Hee University, Seoul 02447, Republic of Korea; sbae@khu.ac.kr

**Keywords:** hospitals, pandemics, physical environment, work environment

## Abstract

Human beings have encountered different infectious diseases. However, there is not much validated data available on the physical environments of hospitals when responding to highly contagious viruses, such as COVID-19. This study was conducted to assess the physical environments of hospitals during the COVID-19 pandemic. There exists a need to analyze whether the physical environments of hospitals were conducive or obstructive to medical practice during the pandemic. A total of forty-six staff working in intensive care units, progressive care units, and emergency rooms were invited to participate in a semi-structured interview. Out of this group, fifteen staff members participated in the interview. They were asked to list the changes made to the hospital’s physical environment during the pandemic, which included equipping the hospital environment for medical practice and protecting staff from becoming infected. They were also asked about desirable improvements that they believe could increase their productivity and ensure safety. The results indicated the difficulty in isolating COVID-19 patients and converting a single occupancy room into a double occupancy room. Isolating COVID-19 patients made it easier for staff to care for the patients, but it made them feel isolated and at the same time increased the walking distance. Signs indicating a COVID area helped them to prepare for medical practices ahead of time. Glass doors provided greater visibility and enabled them to monitor the patients. However, the dividers installed at nursing stations were obstructive. This study suggests that further research should be conducted once the pandemic is over.

## 1. Introduction

Hospitals are complex environments with a lot of information for patients, visitors, and staff. In addition, hospitals are busy and time-sensitive places that deal with urgent situations. Patients fight against their illnesses, and staff take care of them regardless of time or day. Therefore, hospitals can easily become intense and stressful environments [1].

After the advent of COVID-19 in December 2019, the World Health Organization (WHO) declared coronavirus disease a global pandemic. At the time of writing, approximately 364 million cases and over 5.6 million deaths have been reported worldwide [2]. The COVID-19 global pandemic has put healthcare personnel (HP) under immense stress, as they fight against the virus at the frontlines. Healthcare professionals assume critical responsibility and duty, which in turn results in intense pressure and stress [3]. While treating patients at the frontline, they are at the risk of contracting the coronavirus and transmitting it to their loved ones. Unfortunately, in the normal course a pandemic lasts for about two years; hence, the world will not be able to fully overcome the pandemic in the near future. This means that HP should remain cautious and be wary of the virus. At the moment, the pandemic is the new normal and HP are still our real heroes. However, this global pandemic has deeply affected HP’s work performance, as well as their physical and emotional health [4].

Hospitals adopted several cross-cutting strategies in response to the surge in COVID-19 cases, such as creating buffer areas between wards, and placing dividers between contaminated and non-contaminated areas [5]. As a result, HP’s working conditions have changed [6]. For example, hospitals began allowing employees to work from home to minimize the number of people on site, and retraining staff for areas of need [7]. Staff had to adapt to the new work environments and were only allowed to have limited interactions with colleagues owing to social distancing, which resulted in psychological stress and burnout [8].

A lesson learned from Middle East respiratory syndrome (MERS) is that providing a safe and supportive working environment to HP is important to protect them against the virus and prevent the transmission of the virus to others [9]. In this regard, it is crucial for HP to be aware of the mandatory occupational safety and health standards, which include wearing appropriate personal protective equipment and hand-sanitizing. Moreover, it is important to understand the impact of the physical environment on human behavior and perceptions. Hence, this paper explores how HP perceive the hospital environment, and which of its features are conducive, or obstructive, for their medical practices during the COVID-19 pandemic.

## 2. Literature Review

To understand health and disease, a human ecological framework has been frequently adopted, as the framework considers the relationship between humans and their environment based on a theoretical view on health and disease [10]. Based on human ecological perspectives regarding health and disease, there is a growing body of literature on the impact of the physical environment on HP’s satisfaction, productivity, work efficiency, and wellness (i.e., physiological as well as psychological wellness). For example, the spread of infectious diseases can be prevented and/or promoted through various interior environments in medical facilities, and those environments can influence people’s contrasting behaviors on becoming infected [11]. Furthermore, HP’s response to their physical environment is critical, because it directly affects the patient’s health status. For instance, HP’s satisfaction and wellness could affect their productivity, which would have an impact on their work efficiency, potentially resulting in acute reactions toward patients. In addition, burnout due to psychological disorders may cause staff shortages at hospitals. Therefore, providing a supportive environment for HP is important for maintaining stable healthcare environments for both staff and patients. 

Unexpectedly, the COVID-19 pandemic necessitated healthcare facilities to reach out to building occupants. HP are at a higher risk of contracting COVID-19, as they have to take care of infected patients who are in direct contact with diverse fomites, and inactive objects that can carry infection, such as furniture and doorknobs. Although evidence shows that the coronavirus is more contagious via droplets than fomites [12], we cannot overlook the possibility of transmission via fomites. The highly virulent and contagious virus can survive on surgical masks for four (inner layers) to seven (outer layers) days at a temperature of 22 °C (72 °F) and a relative humidity of 65% [13]. Viral particles can stay and suspend on surfaces, and can also travel over 2 m (6 feet) under natural and mechanical airflow [14]. 

Many reports have pointed out that the COVID-19 pandemic escalated HP’s psychological stress and presented a considerable professional burden to them [15]. Unfortunately, what we learned from previous pandemics is that these severe psychological stresses among HP fighting against the virus at the frontline can go on long after the end of a pandemic [16]. Indeed, recent COVID-19 related studies have reported extreme psychological stress, such as isolation and burnout among HP [17]. The stressors include fear of becoming infected and transmitting the virus to others, community perceptions of workloads, shortage of PPE and hospital rooms, and moral dilemmas [18].

Although previous infectious diseases, including SARS, MERS, and Ebola, provide us with valuable lessons to deal with the COVID-19 pandemic [19], more studies are needed to understand the effects of pandemics on HP [6]. Each hospital needed to develop an infection control protocol based on prior epidemics, due to a lack of scientifically validated data [5]. Furthermore, there is much less qualitative research to explore HP’s psychological status during a pandemic [6], compared to quantitative studies [20]. In addition, few studies have examined the role of physical environments during pandemics [21]. Therefore, this paper aims to learn about the aspects that are conducive, as well as obstructive, to HP’s medical practices during the COVID-19 pandemic through their vivid lived experiences.

## 3. Materials and Methods

An explorative study was designed to explore how HP perceived interior environments that facilitated medical practices during the COVID-19 pandemic. Data were collected through qualitative research and semi-structured interviews carried out at Midwest University Hospital in the United States. Three departments—intensive care units (ICUs), progressive care units (PCUs), and emergency rooms (ERs)—participated in the study. These three departments were selected because they treated COVID-19 patients during the pandemic. 

### 3.1. Participants

The study was introduced to the HP working in the three departments via email. To increase the participant rate, the departments’ head sent the invitation email to all HP. As a result of a reminder invitation 10 days after the initial invitation, 46 potential participants showed an interest in participating in this study. Among the 46 participants, a total of 15 HP (four from the ICU, four from the PCU, and seven from the ER) completed the interview. Fourteen of them were female participants. The average age of the participants was 27.75 years, and the age range of the participants was from 21 to 50 years old. Finally, the average number of years in practice was 6.9 years, ranging from 1 year to 30 years. 

### 3.2. Data Collection

This study was approved by the institutional review board of [a university to be named]. Prior to taking part in the interview, all participants read the consent form and verbally agreed to proceed with the interview. 

The interview began with an introductory question concerning the demographic information of the candidates, such as their age, department they worked for and number of years in practice. Later, the participants were asked five major questions. One question was regarding the changes that needed to be implemented in medical practice; two questions pertained to the relationship between the physical environment (room separations, doors, furniture placement, layout, etc.) and medical practice; and the other two questions related to the physical environment and the participant’s perception of safety during the pandemic. Participants shared the aspects that helped, as well as hindered, treatment during the pandemic. They also talked about strategies that would help them improve the environment for better medical practice and prepare them for any future outbreaks. Finally, the participants were asked how the physical environment affected their perception of safety during the pandemic. Furthermore, they were asked about ways to create a better environment to make them feel safe during future outbreaks. 

The questions were developed to explore the participants’ lived experiences and to examine their perceptions of the working environment during the COVID-19 pandemic. During the interviews, the respondents were not prompted to possible responses. However, when the participants’ responses were unclear or ambiguous, they were asked to elaborate further. All interviews were conducted virtually, because the university hospital suspended visitors on site to prevent the spread of the coronavirus. Each interview lasted approximately 30 min and was recorded. Data were collected between February and March of 2021.

### 3.3. Data Analysis

The recorded interviews were transcribed by a professional transcription business using Microsoft Word (Microsoft, Redmond, WA, USA). The transcribed interview responses were analyzed by two researchers to investigate the themes of HP’s lived experiences during the COVID-19 pandemic. The responses were individually coded for primary and secondary themes using Microsoft Excel (Microsoft, Redmond, WA, USA). The two researchers independently coded the interviews, and then compared their coding to check whether they had coded the interviews in the same way. Following the analysis, the themes were gradually developed, based on discussions to enhance rigor, and it was found that the primary themes were broader than the secondary themes. An answer with more than one theme was coded into each different theme. Regarding the unconformable themes, the researchers had a discussion to hear each other’s rationales, before reaching agreements on those themes. The overall inter-rater reliability was found to be 0.89. 

## 4. Results and Discussion

### 4.1. Changes Regarding Medical Practice

The patients’ rooms, PPE and equipment across the three units underwent certain changes after the outbreak. Specifically, the patients with respiratory and non-respiratory issues were separated from one another to prevent the spread of the virus within the unit. In addition, the PCU began offering ICU level care, because of the large number of COVID-19 patients. Owing to room shortages, at times they had to convert single occupancy rooms to double occupancy rooms, which meant that double the amount of equipment was needed for treatment. This resulted in the rooms becoming very crowded, while at the same time staff experienced a significant increase in workload. One PCU staff member (31-years-old, 10–15 years in practice) recalled that “*As the numbers increased, we started taking more ICU overflows for these patients. … Most of our rooms are already set up for ICU level of care, but we prepped our rooms for like double occupancy, and then actually had double occupancy. That was also a big change*”.

Staff also pointed out the changes that were implemented in PPE and equipment. At the beginning of the pandemic, there was a shortage of PPE. The unit decided to put intravenous (IV) poles outside the patients’ rooms, as they faced a shortage of PPE and space constraints because of more than one patient and the equipment used to care for them being in the rooms. This way, staff did not need to put on full PPE to be able to access the room, and they were also able to minimize the clutter of cords from the IV poles and equipment. One staff member working in the ICU (unspecified age, more than 30 years in practice) mentioned that, “*The IV poles would be outside of door. So nurses did not have to put on a full PPE to be able to get in and [could] just turn off an alarm or readjust a rate on a drip or anything like that*”.

### 4.2. Physical Environments Affecting Medical Practice

Ten of the 15 participants opined that the equipment aided and at the same time hindered their medical practice during COVID-19. The positive aspect was that it ensured accessibility to medical supplies. Prior to the pandemic, they had medical carts with basic supplies in each room, so as to not frequent the supply rooms. They also had to carry additional items to treat COVID-19 patients more effectively during the pandemic. However, additional equipment crowded the patients’ rooms. One ICU staff member (unspecified age, more than 30 years in practice) recalled that, *“In many ways, it made things more difficult. These patients have been more critically ill than I’ve ever seen. So, they require more equipment in the room and more staff to take care of these patients. So, the room was constantly too small”*.

In addition, separating COVID-19 patients from the other patients aided medical practice. The hospital had two places where patients could check-in, depending on their symptoms. *“We separated [the rooms] into pods so we have one COVID pod and we have doors erected and that was only supposed to be [for] the respiratory COVID patients and that was very helpful.”* (ER staff member, 23-years-old, less than 5 years in practice) Sometimes, patients had a longer wait time in a designated area, but staff mentioned that this was to protect patients from being exposed to COVID-19 patients. This method helped staff with their workflow. Information regarding the layouts that they would be working with each day enabled them to focus on the COVID-19 patients. Furthermore, the signage was extremely helpful. Staff posted a “COVID RESPONSE AREA” sign on the doors that led to the COVID-19 pod, and attached a red flag or made the room monitor red to indicate the presence of infected patients in a certain area (Figure 1).

However, staff had to prepare double-occupancy rooms whenever there was a surge in COVID-19 patients. They had to remove a lot of furniture, from couches to bedside tables, for patients, since the rooms were originally designed for single-occupancy. A staff member (PCU staff, 21-years-old, less than 5 years in practice) recalled that they needed to work extra hard, and it was stressful for both patients and staff. *“It was kind of scary when a patient would be in a bad situation and they had a roommate. I would have to calm both of them down at once and maintain some kind of calmness out of all [of] it, which was never easy. So, the double bed was definitely the hardest part of it then.”* Furthermore, the COVID-19 visitation restrictions imposed by the hospital had a negative impact on the patients, as they were not receiving social support. 

Additionally, most participants did not like the plexiglass installed between workstations to prevent the spread of coronavirus (Figure 2). They complained that the barriers installed in the middle of the ledge caused spatial constraints at the nursing stations. An ER staff member (23-years-old, less than 5 years in practice) mentioned that, *“Something that hindered us all are these erections, like the glass wall for no reason…. What is the difference between putting up a glass wall between me and my coworkers and going into a room and then flipping a patient together [by] standing right next to each other?”* On the other hand, staff liked the barrier between patients and staff, especially in a COVID pod. 

Lastly, glass doors enabled greater visibility, as staff could see their patients without having to wear PPE to go in. All the rooms could be monitored from the ICU nursing station and ICU staff appreciated the high level of visibility. 

### 4.3. Physical Environments to Be Improved for Better Medical Practices

Most of staff mentioned the need to improve the physical environment. For instance, the travel path was elongated, as the COVID-19 patients of the corner pod had to be separated from the non-COVID patients. This negatively affected the workflow. Staff mentioned that the isolated COVID pod with the most critical patients was located far from the major medical resources and healthcare team. A staff member (ICU staff, 26-years-old, 6–10 years in practice) particularly noted that, “*Having staff in an isolated pod makes getting resources harder, and prompt response to critical situation is lower because additional aid comes from opposite side of department.*” Physical fatigue and emotional depletion have also been reported. One ER staff member (31-years-old, 11–15 years in practice) stated that, *“I liked that we separated the [patients with] respiratory illness to one pod and I think that makes sense for patient care. However, I think it can create a confidentiality issue and a nurse burnout issue if nurses are not moved around to different pods.”* Furthermore, staff wanted to have spacious rooms to house more equipment, and negative pressure rooms to treat patients with respiratory diseases. They also wanted to either reorient the rooms or retain the same layout throughout the unit. 

Additionally, staff wanted to improve the technology and equipment. Since, patients’ rooms contain many pieces of equipment, a lot of electrical cords often lie across the floor. At times, staff are unable to tuck them away safely. Therefore, they suggested that advanced technologies, such as wireless devices, lift equipment, and outlets on the ceiling or the floor near the patient or their bed, would make it much easier for them to take care of the patients. They also proposed the use of advanced communication methods, such as a call light to indicate emergency and urgent needs. One PCU staff member (21-years-old, less than 5 years in practice) mentioned that, *“Knowing which rooms need the most attention because you have to gown up and may waste PPE. It takes 10 times longer to get in and out of a room. … so I think definitely having [a] better way of communicating what each room needs and which one on a higher basis need would be very helpful.”* In addition to the need for advanced technology and equipment, staff would like to improve the availability and accessibility of equipment, as it is important for critical patients. A staff member (PCU staff, 50-years-old, 6–10 years in practice) pointed out that, *“Access to certain equipment is limited because [they are] not always being placed in the most efficient way for access on COVID side to help limit nurse from walking throughout the department when in a very sick COVID room”*.

### 4.4. Impact of Physical Environments on COVID-19 Safety Perceptions

Above all, appropriate and sufficient PPE and equipment made staff feel protected from COVID-19. At the very beginning of the pandemic, the shortage of PPE was a major issue that made them feel extremely unsafe. An ER staff member (22-years-old, less than 5 years in practice) stated that, *“In the beginning, due to the shortage, we were reusing stuff that we shouldn’t have been reusing, to be honest. However, that is all we could do. So now things are improved and in terms of feeling safe, I pretty much do feel safe at work.”* Easy access and organization are critical. At times, PPE and equipment carts are not properly stored and instead left out in the hallways. This makes it easier for staff to find the things they need. The carts might cause crowding of the hallways; however, it does not directly impede movement. A staff member (ICU staff, unspecified age, more than 30 years in practice) said, “We *do have the ability to line equipment up even outside of our doors that still allow people to be able to walk around. It does get crowded, but, you know, at least we have that ability that we can put the equipment right outside the doors and still function”*.

Glass doors also provided greater visibility and made staff feel safe. A staff member (ICU staff, 26-years-old, 6–10 years in practice) noted that, *“The glass doors were amazing and super helpful, and like the way you can see in the rooms from the nursing station made me feel safe. We have the ability to close those so that definitely has made things safer.”* In addition, signage on doors indicating a COVID-19 pod or a COVID-19 patient’s room made them feel safe, as it indicates the need for caution. Despite the limited number of negative pressure rooms, staff appreciated having these rooms in the unit, as it made them feel safer. Furthermore, one PCU staff member (21-years-old, less than 5 years in practice) pointed out that the PCU department has spacious rooms compared to the other units. *“I love our floor, especially in PCU is on the newer part of the hospital and it has a lot bigger rooms usually because the patients have more family that want to come in. But, since COVID, the bigger rooms have definitely helped, and I felt much safer”*.

There was some negative feedback regarding isolating COVID-19 patients, as this can lead to emotional detachment. However, most staff felt safer because the isolated COVID-19 units were provided with extra PPE, including N95 masks. In addition, staff felt less safe after they started double-bedding, because of limited ventilation and double occupancies in a single room. They also noticed that the health status of the roommate significantly affected the patient’s health status. 

### 4.5. Physical Environments to Be Improved for Better Perceptions of Safety during COVID-19

Staff wished to improve the cleanliness of the rooms, as well as their ventilation systems. For instance, in the ER, staff have to flip the routes quickly, so that they can wipe down the main surfaces. Staff felt that it was not very clean. One ER staff member (22-years-old, less than 5 years in practice) mentioned that, *“In terms of cleanliness, it would be great to mop the floors, clean the glass, wipe patient doors as those are always smeared and dirty and I hardly ever seen them being cleaned. I also think that we need to pay more attention to the glass at the nurse station. I guess there’s always [scope for] improvement.”* In addition, one staff member suggested cleaning the room with UV light. Staff opined that they would feel safer if the patients’ rooms had proper a ventilation system for viruses, particularly when they needed to create double-occupancy rooms. 

Moreover, staff wanted a more durable divider (i.e., a large plastic panel) to be placed between the patients in the double occupancy rooms. The lightweight divider that was being used was frequently knocked over, especially when staff were helping patients or shifting things around. The divider poses a danger to the patients and can also damage critical equipment (Figure 3).

Lastly, there were comments regarding the need to replace the hand sanitizers. A staff member (PCU staff, 31-years-old, 11–15 years in practice) said that the sanitizer that is being used has a bad odor and makes their hands dry, thus resulting in a low hand-hygiene compliance rate. The staff member said, *“I am very grateful for all the hand sanitizer that was given but it smells so bad. Therefore, we noticed that it could have decreased compliance because of the scent. Over the top of it, when we take our gloves off, oh my gosh! It just smells even worse!”*

### 4.6. General Discussion

Through interviews, participants shared their thoughts based on their lived experiences, regarding various aspects of the hospital environment that are conducive, as well as obstructive, to medical practice during the COVID-19 pandemic. The recruited hospital separated COVID-19 patients from non-COVID-19 patients. Most participants felt that isolation helped improve the workflow, as they could focus solely on COVID-19 patients if they were assigned to designated areas. The designated space for COVID-19 patients helped minimize the walking distance between the wards. 

Researchers have suggested that isolating infectious patients is one of the key design strategies for resilient hospitals to limit cross-contamination [5]. In particular, spatial separation and negative pressure isolation spaces help HP in controlling transmission [22]. However, these measures sometimes cause limited social interactions among colleagues [8]. A systematic review of social support among hospital staff during the COVID-19 pandemic described communication with colleagues, family, and friends as primary coping skills for staff to manage their mental health [23]. Furthermore, this study found that isolating COVID-19 patients could increase HP’s walking distance. This may lead to walking fatigue, excessive time spent in walking instead of taking care of patients, and delayed medical care [24]. 

Other helpful features during the pandemic were intuitive signage and aspects that enhanced visibility. Signage indicating designated COVID-19 areas helped staff prepare themselves mentally and physically to take care of the patients. In addition, high level of visibility though glass doors allowed them to know what was happening on the other side of the ward or in the room. The participants mentioned that the ability to see the inside of the ward was extremely helpful, especially as they did not have to put on the appropriate PPE to check on the patients. 

Research has shown that a high level of visibility enhances staff awareness and team communication [25]. During the pandemic, hospitals put up signage, called the ‘COVID-19 flag’, which provided a quick visual guide [26]. Even though many studies have reported the benefits of good wayfinding signs in hospitals [27], more studies are needed to establish the scientifically validated benefit of an intuitive signage for caution.

In addition, the results indicate the importance of appropriate medical equipment to treat patients and the need to ensure its adequate supply Moreover, participants also wanted easy access to medical equipment and/or supplies, for better work productivity, as it would help reduce walking distances. 

Previous literature implies that standard locations for equipment and supplies can enhance patient and staff outcomes as part of evidence-based design strategies [28]. To establish standard locations, space should be allocated for large equipment, as well as additional equipment, if required [29]. Furthermore, an important lesson learned from the COVID-19 pandemic is that, while designing the room, account must be taken of the space available to accommodate extra equipment for emergency situations. 

During a pandemic, hospitals experience room shortages. For controlling infectious diseases, single occupancy rooms have more advantages than double occupancy rooms [30]. Despite the benefits, many hospitals, including the hospital assigned for this study, had to convert single occupancy to double occupancy rooms, due to a surge in COVID-19 patients. During the pandemic, double occupancy rooms became crowded with extra beds and medical equipment, and it was difficult for HP to take care of two patients at the same time. The design of the rooms must be such that it allows for rapid conversion into double occupancy rooms, with adequate space for electric outlets and equipment. 

Lastly, dividers and barriers were helpful in preventing the transmission of the virus, but they have to be made sturdy. To be specific, dividers between staff and patients, or between patients, were helpful; however, dividers between staff at nursing stations were obstructive, as they occupied too much space on the desks. In addition, dividers that aided in converting single occupancy rooms into double occupancy rooms must be durable. Unfortunately, we were unable to learn about dividers and barriers from previous pandemics. In the future, durable dividers must be installed in patients’ rooms, where communication between patients and staff occur, rather than in nursing stations.

## 5. Limitations

One of the limitations of this study was that the participants volunteered for it. As the sample was not representative of the hospital, the findings cannot be generalized. Another limitation is that the participants’ thoughts may have changed after data collection. Data were collected immediately after the second wave (January 2021) of the pandemic in the United States. The pandemic has not yet been overcome. Over the past two years, with different variants of the virus (e.g., Delta and Omicron), both the conducive and obstructive features mentioned by the participants may have changed. In addition, fourteen out of the fifteen participants were female; hence, the results cannot be generalized. Therefore, further study must be conducted once the pandemic is over to garner insights regarding the physical environments in hospitals during the COVID-19 pandemic. 

## 6. Conclusions

The unpredictable COVID-19 pandemic struck the world. All institutions, such as businesses and educational establishments, have been put under pressure. However, hospitals face a greater degree of stress than any other type of institution. Scholars have conducted investigations on the important role of the physical environment in hospitals in preventing the spread of infectious diseases. However, not much study has been carried out to understand the role played by the physical environment during a pandemic. Hence, we need to rely on the lessons learned from previous pandemics, such as SARS and Ebola, which are much less contagious than COVID-19. This study examines the interior environments that were conducive, as well as obstructive, for hospital staff during the pandemic. 

Separating COVID patients from non-COVID patients was helpful for staff as it enabled them to prepare mentally and physically to take care of COVID patients. However, they felt isolated from their colleagues owing to the lack of communication. In addition, single occupancy rooms were converted to double occupancy rooms to treat COVID patients. This was challenging for staff, as they had to take care of two patients at the same time. The rooms became crowded with additional equipment for the second patient. Therefore, in future, the rooms must be designed in such a way that the patients have adequate space. Furthermore, the design of rooms must be such that it allows for rapid conversion into double occupancy rooms. As aforementioned, staff found the signage for designated COVID-19 areas very helpful. Glass doors offered them increased visibility and allowed them to monitor the ward, as well as the patients’ rooms. This in turn prevented wastage of PPE suits. Sturdy dividers are necessary to convert a single occupancy room into a double occupancy room. Dividers at nursing stations were obstructive for staff, as they occupied space on the desks. However, the dividers between the patients and staff made them feel safe from infection. 

These painful lessons could be a starting point for fighting against a global pandemic in the future. Insights garnered from this pandemic will aid in providing a more supportive and safe hospital environment for our heroes. 

## Figures and Tables

**Figure 1 ijerph-20-03271-f001:**
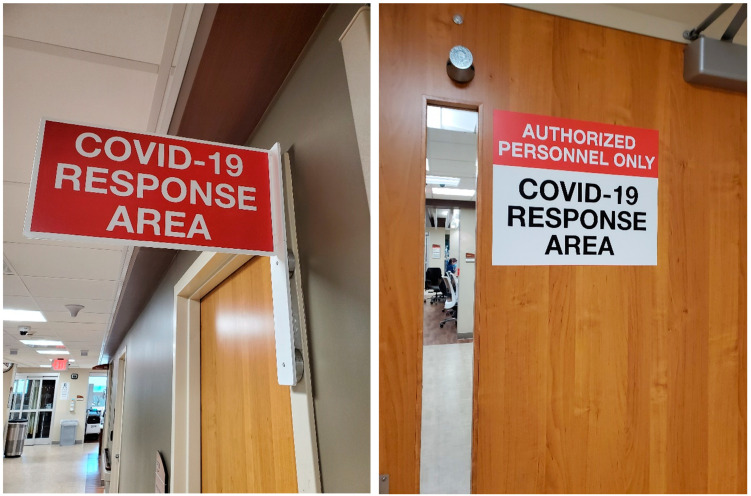
“COVID-19 RESPONSE AREA” signage on the doors.

**Figure 2 ijerph-20-03271-f002:**
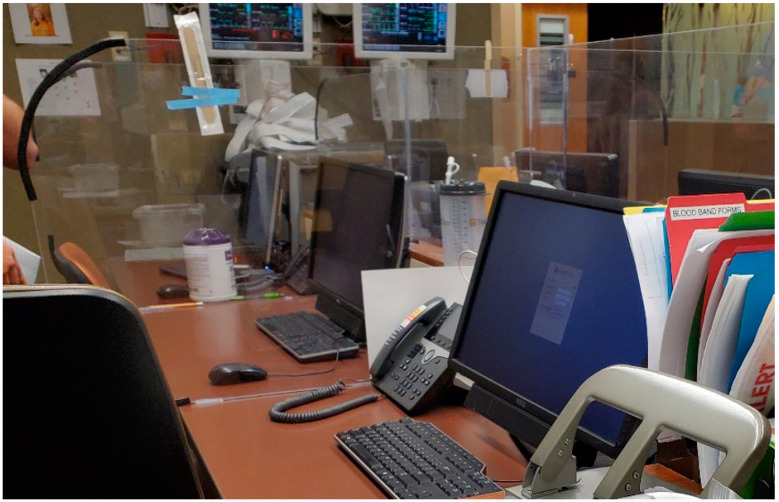
Dividers installed between workstations.

**Figure 3 ijerph-20-03271-f003:**
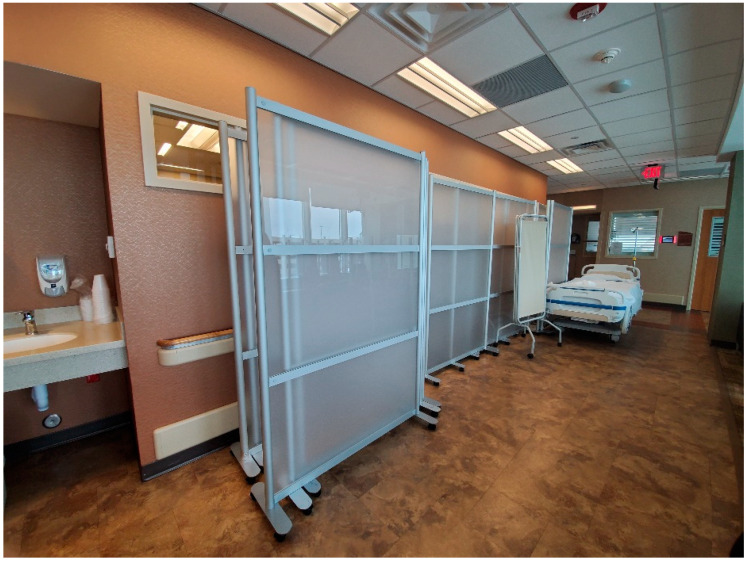
A divider placed between the patients in the double occupancy rooms.

## Data Availability

The data is not available because of confidentiality.

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
