# Peer review of "A Qualitative Study of Hospital Interior Environments during the COVID-19 Pandemic"

_ijerph, 2023, doi:10.3390/ijerph20043271_

Round 1
Reviewer 1 Report
1. More references on safety and wealth control can be mentioned in the Introduction, especially those were conducted in the context of medical facilities.
2. The main argument / theory support has not yet been highlighted in the Introduction. If there is no literature review section, a cohesive summery of existing literature and theoretical framework / analytic framework to support research design should be provided.
3. The biggest weakness of this paper lies in the methodology. I believe that the study adopted a solid methodology, but it is not well illustrated in this paper, such as the sampling process, the profile of the interviewees, etc. Also, the data analysis is vaguely mentioned. This has undermined the validity of the results and discussion.
4. What are the differences between "general discussion" and other sections under "discussion"?
Author Response
Dear Editors and Reviewers,
First of all, thank you very much for your thoughtful feedback. All the comments were very helpful to improve our manuscripts. We updated our manuscript based on your comments. Please see our detailed responses to each comment. Red text indicates our responses to reviewers’ comments. I hope you can see improvements in the revised manuscript. If you still think we need to improve our manuscript to publish, please let us know. We are happy to hear your opinions and update our manuscript to make it stronger. Also, if you have any questions, please let me know.
I truly appreciate your time.
|
Reviewers’ Comments |
Authors’ Responses |
|
Reviewer #1 |
|
|
1. More references on safety and wealth control can be mentioned in the Introduction, especially those were conducted in the context of medical facilities. |
[Pages 3-4] Thank you very much for your insight. I put a paragraph in Introduction to explain more about safety and wealth control in hospitals, especially during the COVID pandemic.
I also added one more sentence about relationship between interior environments and safety in Literature Review. Please let me know if you think the addition is not enough. Thank you! |
|
2. The main argument / theory support has not yet been highlighted in the Introduction. If there is no literature review section, a cohesive summery of existing literature and theoretical framework / analytic framework to support research design should be provided. |
[Page 4] Based on your comment, I separated Introduction and Literature Review. Please see page 4. I also specified a theoretical framework (human ecological framework) at the very beginning of Literature Review.
Thank you very much for your great insight on the manuscript. |
|
3. The biggest weakness of this paper lies in the methodology. I believe that the study adopted a solid methodology, but it is not well illustrated in this paper, such as the sampling process, the profile of the interviewees, etc. Also, the data analysis is vaguely mentioned. This has undermined the validity of the results and discussion. |
[Pages 6-7] This is a good point. I added more information about recruitment process and profile of the interviewees.
[Page 8] In addition, the data analysis part was improved. |
|
4. What are the differences between "general discussion" and other sections under "discussion"? |
[Page 16] “6. General Discussion” on p. 16 is about the overall discussion above the points, from 1 to 5 discussion points. If the word is confusing, I can change it to “Overall Discussion”. Please let me know what you think. Thank you! |
|
Reviewer #2 |
|
|
Participants: Indicate also the Age Range of participants. Is there also data on the contraction of Covid-19 by the participants? Also, it could be interesting to know if people with a long experience reported different opinions from healthcare professionals with less experience. |
[Pages 6-7] Thank you very much for your great feedback. The age rage information is included. Also, work years are included. Unfortunately, this study only asked a minimum of demographic information about participants; therefore, there is no more information about participants to be reported. |
|
Data analysis: Please, improve this section. |
[Page 8] The data analysis part was improved. |
|
Results: When reporting excerpts from their interviews, please indicate age, medical unit, and years of service. This will add richness to the reported content. |
[Pages 8-16] This is also great point. Now, age, medical unit, and years of practice were indicated. Regarding the years of practice, I specified as a range (such as less than 5 years, more than 30 years) because the participants can be revealed due to the small sample size.
If you still think that exact service years should be identified, please let me know. Happy to revise it. Thank you! |
|
Also, have you considered a results' graphic representation? |
This is a great point. I thought about to add a graphic representation. However, due to the small sample size and nature of the qualitative data, I ended up including pictures to show visually what the participants tried to explain. I hope this makes sense to you. If there is anything I can improve the manuscript, please let me know. Happy to hear your opinions. |
|
Limitation: Add the almost total presence of the female gender. |
[Page 19] Thank you for the great point. I added the gender as a limitation at the end. I appreciate your insights on the manuscript. |

Reviewer 2 Report
Wellbeing of healthcare professionals after the advent of Covid-19 pandemic is an important topic to be explored. The authors focused on the physical features of three different medical units and their influence on the medical practice through a qualitative study. However, some minor issues should be addressed. My comments are as follows:
Participants: Indicate also the Age Range of participants. Is there also data on the contraction of Covid-19 by the participants? Also, it could be interesting to know if people with a long experience reported different opinions from healthcare professionals with less experience.
Data analysis: Please, improve this section.
Results: When reporting excerpts from their interviews, please indicate age, medical unit, and years of service. This will add richness to the reported content.
Also, have you considered a results' graphic representation?
Limitation: Add the almost total presence of the female gender.
Author Response

(The authors gave the same response as above.)

Round 2
